# Sickness absence rates in NHS England staff during the COVID-19 pandemic: Insights from multivariate regression and time series modelling

Ewan McTaggart[1]*, Itamar Megiddo[2], John Bowers[3], Adam Kleczkowski[1]

1 Department of Mathematics and Statistics, University of Strathclyde, Glasgow, United Kingdom,
2 Department of Management Science, University of Strathclyde, Glasgow, United Kingdom, 3 Stirling Management School, University of Stirling, Stirling, United Kingdom

* ewan.mctaggart@strath.ac.uk

## Abstract

The COVID-19 pandemic placed immense strain on healthcare systems worldwide, with NHS England facing substantial challenges in managing staff illness-related absences amid surging treatment demands. Understanding the impact of the pandemic on sickness absence rates among NHS England staff is crucial to developing effective workforce management strategies and ensuring the continued delivery of healthcare. In this study, we use publicly available data to investigate the impact of the COVID-19 pandemic on sickness absence rates among NHS England staff between June 2020 and 2022. We begin with a data analysis to indicate the temporal patterns of sickness absence in NHS England staff between January 2015 and September 2022 inclusive. We then develop multivariate linear regression models to estimate COVID-19-related sickness absences. Indicators of COVID-19 activity, such as positive tests, hospitalizations, and ONS incidence, were incorporated. Furthermore, we use Seasonal ARIMA time series models to analyse the impact of COVID-19 on mental health-related absence. Our analysis highlights increases in sickness absence rates which coincide with the arrival of COVID-19 in England, and continue to rise throughout the pandemic. High periods of COVID-19 activity strongly correlated with staff absence, and the main categories driving the dynamics were COVID-19-related or mental health absences. We demonstrate that sickness absences in these two categories can be estimated accurately using multivariate linear regression (F(2, 15) = 132.63, $P < .001$, adj $R^2$ =93.9%) and Seasonal ARIMA time series models, respectively. Moreover, we show that additional indicators of COVID-19 activity (positive tests, hospitalisations, ONS incidence) contain helpful information about staff infection pathways. This study offers insights into the dynamics of healthcare staff absences during a pandemic, contributing to both practical workforce management and academic research. The findings highlight the need for tailored approaches to address both infectious disease-related and mental

**Data availability statement:** All data underlying the findings described in this manuscript are fully available without restriction. The sickness absence data were obtained from the NHS England Sickness Absence Rates publication and are accessible at https://digital.nhs.uk/data-and-information/publications/statistical/nhs-sickness-absence-rates. COVID-19 surveillance data were sourced from publicly available repositories, including the Office for National Statistics (ONS) Coronavirus (COVID-19) Infection Survey, available at https://www.ons.gov.uk/peoplepopulationandcommunity/healthandsocialcare/conditionsanddiseases/bulletins/coronaviruscovid19infectionsurveypilot/previousReleases, and the UK Health Security Agency (UKHSA) Coronavirus Dashboard, accessible at https://ukhsa-dashboard.data.gov.uk/respiratory-viruses/covid-19. All code used in the paper is available on GitHub: https://github.com/ewanmct/COVID_NHS_ABSENCES.

**Funding:** This study was financially supported by the University of Strathclyde in the form of a Student Excellence Award received by EM. No additional external funding was received for this study. The funder had no role in study design, data collection and analysis, decision to publish, or preparation of the manuscript.

**Competing interests:** The authors have declared that no competing interests exist.

**Abbreviations:** AIC, akaike information criterion; AR, autoregressive; COVID-19, coronavirus disease 2019; FTE, full-time equivalent; HCW, healthcare workers; MA, moving average; NHS, National Health Service; ONS, Office for National Statistics; PCR, polymerase chain reaction; SARIMA, seasonal autoregressive integrated moving average; UK, United Kingdom; VIF, variance inflation factor

health-related absences in healthcare settings during future health crises and opens new avenues for research into healthcare system resilience during crises.

## Introduction

Sickness absence is a substantial social and economic burden to the National Health Service (NHS) England, costing an estimated £1.65 billion each year [1]. NHS England is the largest employer in the UK, with 1.2m full-time equivalent staff as of April 2022 [2], but their monthly average sickness absence rate far exceeds the UK public sector (4.2% in NHS staff [1] vs 2.9% [3], between 2009–2019). This absenteeism can hamper the provision of patient care, strain the working conditions of the remaining staff, and incurs expenses for sick pay and temporary staffing [4].

Healthcare workers (HCWs) have historically faced an increased risk of occupational infection [5] during epidemics, and the COVID-19 pandemic was no exception [5]. Patient-facing healthcare workers in NHS Scotland were three times more likely to be hospitalised with COVID-19 compared to the general population [6]. Appelby et al. [7] identified a large excess sickness absence in NHS England staff during March-May 2020 compared to the previous ten-year average for each month, coinciding with the first wave of the pandemic. Furthermore, COVID-19 outbreaks recurred in waves outwith the winter seasonal pattern that NHS was used to with influenza [8]. These waves exhibited multiple peaks and high troughs, resulting in constant strain on the system with few lulls. During these waves of COVID-19, NHS England faced a twofold crisis: surges in treatment demand and staff absenteeism. Reports surfaced of insufficient staff to provide adequate care, placing immense pressure on the system [9]. Although the introduction of vaccines in early 2021 reduced the probability of severe illness [10], the risk of staff contracting disease remained, rendering them unable to work [11].

Emerging evidence suggests the burden on NHS England staff led to absences due to reasons that were not directly COVID-19-related, such as other respiratory diseases and mental health. In a comparison between 2019 and 2020, Edge et al. [12] found increases (at least initially) in asthma, chest and respiratory disease, infectious diseases and mental illness, but decreases in other categories such as musculoskeletal disorders, injury and fracture, gastrointestinal disease, genitourinary and gynaecological disease and, most notably, cancer. Van der Plaat et al. [13] explored absences in NHS England due to mental health over the same period and found a spike in absences in March-April 2020, which declined to typical levels by May and June. They also found regional correlation between the percentage change in new mental health absences and absences attributed to COVID-19, suggesting an interaction between the two.

Before the COVID-19 pandemic, models were developed to predict sickness absence rates while including the effects of seasonal and pandemic influenza. Asghar et al. [14] showed that SARIMA time series models could accurately make 6-month predictions of the overall sickness absence rate in NHS England ambulance staff. In particular, using multivariate regression models to estimate sickness absence

attributed to influenza (seasonal and pandemic) from proxy variables for influenza activity (e.g., hospitalisations or tests) is an established approach in the literature [15–17]. For example, Ip et al. [15] highlighted how periods of heightened influenza activity between 2004–2009 were linked to increased HCW sickness absence in Hong Kong.

Building upon the research by Appleby et al. [7], Edge et al. [12], and Plaat et al. [13], who explored 2019−2020 data, we investigate the impact of COVID-19 on sickness absence rates in NHS England from late 2020 into 2022. In this paper, we develop multivariate regression and time series models using publicly available data on sickness absence rates in NHS England between January 2015 and September 2022 inclusive. Our research addresses several key questions. Firstly, what were the key trends and sources of variability in NHS England Sickness absence rates during this period? Secondly, can we use this information to develop models to explain the sickness absence rates? Lastly, what is the relationship between indicators of COVID-19 activity (e.g., PCR tests) and these sickness absence rates? Understanding these sickness absence trends and future workforce availability is critical for effective healthcare staff and resource planning [18].

## Methods

### Data sources and data

We obtained data on absences in NHS England staff from the NHS England's Sickness Absence Rates publication. This contains monthly observations of sickness absence rate broken down by reason from January 2015 until September 2022, taken directly from the electronic staff record [2]. The data are aggregated at the national level and do not contain individual records or personal information. No additional spatial, demographic, or socioeconomic information is available. The sickness absence rate is defined as the ratio between the "full-time equivalent (FTE) number of days lost" and "FTE number of days available". We define COVID-19-related sickness absence similarly to a previous study [19]: absence in any of three diagnostic categories/reasons; S13 cold/cough/flu, S15 chest and respiratory problems, and S27 infectious disease. Mental health-related sickness absence was recorded under the category S10 anxiety/stress/depression/other psychiatric illnesses.

Additionally, we collected data on four indicators of monthly COVID-19 severity in England: community incidence, positivity, new PCR positive tests, and new hospital admissions. We use COVID-19 community incidence and positivity estimates from the Office for National Statistics Coronavirus (COVID-19) Infection Survey [20]. The incidence rate estimates the number of new PCR-positive infections per day, and the positivity is the number of people who would test positive each day. They reflect COVID-19 infections of people living in private households (general population households and households of NHS staff) but not patients in hospitals or care homes. The daily confirmed PCR-positive COVID-19 tests reported for England were taken from the UK Health Security Agency (UKHSA) Coronavirus Dashboard [21]. The daily number of new patients admitted to hospitals with COVID-19 in England is from the same source. This data counts people admitted to hospitals who tested positive for COVID-19 14 days before admission or during their stay in the hospital. Inpatients diagnosed with COVID-19 after admission are reported as being admitted the day before their diagnosis [22]. We converted all surveillance data to monthly observations for comparability to the sickness absence data. The bi-weekly ONS incidence estimates were first interpolated to get daily estimates, then summed for a monthly one. Similarly, we estimated the average number of people testing positive daily in a given month by interpolating the bi-weekly positivity estimates and taking an average. July 2020 and July 2022 are the first and last months with an observation from all four sources of COVID-19 surveillance data, Fig 1.

### Ethics statement

This study exclusively uses publicly available, anonymised, and aggregated data from official sources. The sickness absence data is published by NHS England and extracted from their Electronic Staff Record in accordance with their data security and confidentiality policies. The data contain no individual identifiers, personal information, or demographic

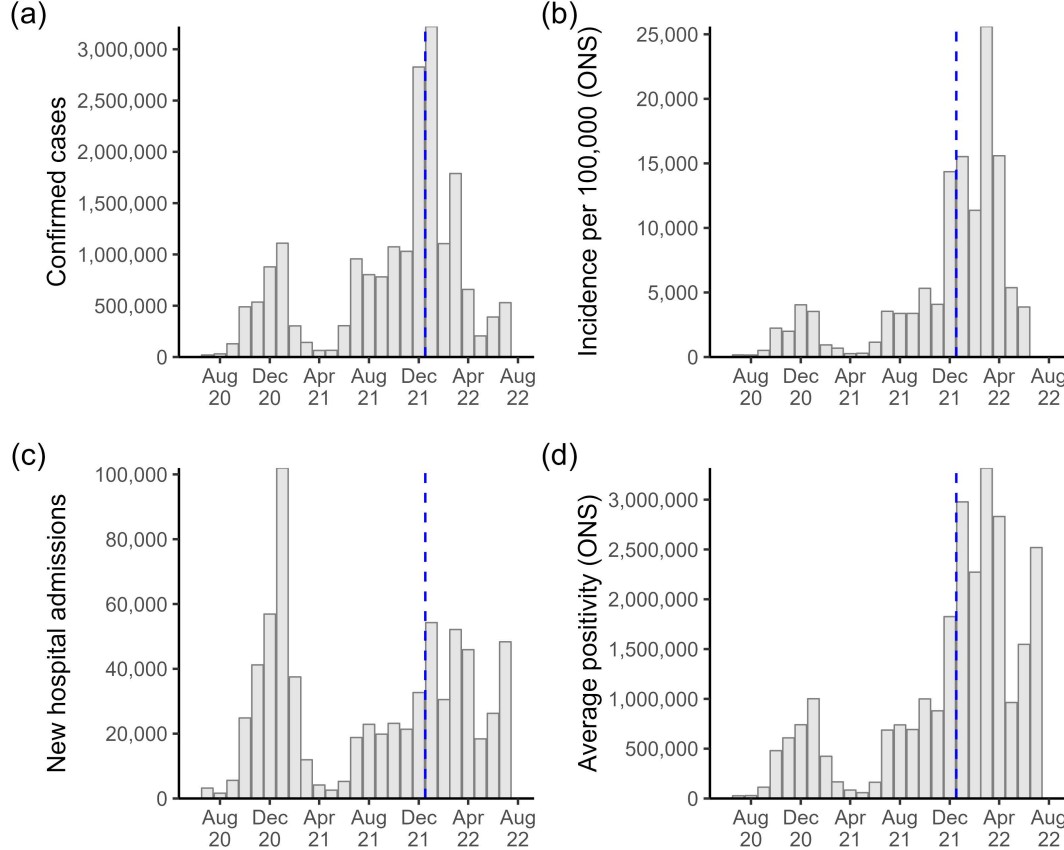

**Fig 1. COVID-19 surveillance data for England.** Monthly observations where months before the dashed blue vertical line were used for developing the regression models. (a) confirmed PCR-positive COVID-19 tests; (b) ONS estimated new COVID-19 infections per 100,000; (c) new COVID-19 hospitalisations; (d) ONS average testing positive for COVID-19 each day. Note that ONS data were not published before July 2020 and the ONS stopped publishing incidence estimates after July 2022 [20].

details. COVID-19 surveillance data were obtained from publicly accessible repositories, including the ONS Coronavirus (COVID-19) Infection Survey and the UKHSA Coronavirus Dashboard, both of which provide aggregated, non-identifiable data.

As all data sources are fully anonymised and publicly available, this study did not require approval from an institutional review board (IRB) or ethics committee. Individual consent was not applicable, as the data contains no personal or identifiable information.

## Statistical analysis

**COVID-19-related absence: Linear regression models.** Multivariate regression can estimate each predictor's independent effect after controlling for confounding factors between predictors. Therefore, to investigate the relationship between indicators of COVID-19 activity and COVID-19-related sickness absence rates, we developed univariate and multivariate linear regression models [23] from all combinations of COVID-19 surveillance data through an exhaustive selection process. Statistical significance was determined by considering a type I error probability below 5% ($\alpha < 0.05$). We compare the fits of these models and their predictive power using adjusted $R^2$ and Akaike Information Criterion (AIC). We included the following covariates: the number of PCR-positive COVID-19 tests, new COVID-19 hospitalisations, and

the community incidence of COVID-19. We expected these indicators of COVID-19 severity to correlate positively with COVID-19-related sickness absence. We train the regression coefficients for the models using data from July 2020 until December 2021. Using the trained models, we estimate the COVID-19 sickness absence trend between January 2022 and July 2022 for comparison to the observed data.

**Mental health-related absence: Time series models.** We used Seasonal Autoregressive Integrated Moving Average (SARIMA) time series models to explore the impact of COVID-19 on mental health-related sickness absence, due to their ability to account for trends, seasonality, and autocorrelation in the data. These models describe the relationship between the current value of a time series and its past values as well as a random error term [24]. They use a combination of auto-regressive (AR, how the current value depends on past values) and moving average (MA, how past errors affect future values) models, as well as differencing (to make the data stationary), to capture trends and seasonality in data [20].

We partitioned our temporal data into different phases of the COVID-19 pandemic and performed a residual analysis, to identify shifts and anomalies in absence patterns. Specifically, we fit one SARIMA model trained on pre-COVID-19 (January 2015-March 2019) data and used it to predict the post-COVID-19 trend. Additionally, we fit a second SARIMA model including some post-COVID-19 months (January 2015-December 2021). We then forecast the absence rates for the first six months of 2022 using these models and compared the estimates to the first six months of observed data. Model parameters were selected to minimise the Akaike Information Criterion (AIC), using the auto.arima function from R's "tim-etk" package [20], with stepwise=FALSE to perform an exhaustive search across all candidate SARIMA(p,d,q)(P,D,Q)[m] model configurations [20]. The resulting fitted models were SARIMA(0,1,3)(0,1,0) [12] for the model using pre-COVID-19 period data for training and SARIMA(2,0,0)(0,1,1) [12] with drift for the model including some post-COVID-19 months. The equations and parameter estimates for both models are given in Multimedia Appendix 1: SARIMA model equations.

## Results

We begin with a high-level exploration of the sickness absence trends between 2015 and 2022, before investigating the reasons for absence in further detail. We then assess the relationship between COVID-19 and sickness absence rates using regression models built from COVID-19 surveillance data. Furthermore, we analyse mental health-related absence using SARIMA time series models, similar to Asghar et al. [12], to highlight the impact of COVID-19.

### Workforce level absence trends

Absences differed substantially in the years following 2020 compared to 2015–2019, and this was driven primarily by absences in the COVID-19-related and mental health-related categories. During 2015–2019 the overall sickness absence rate for NHS England followed a seasonal trend, with peaks in winter (January/December) and troughs in summer (May/June), Fig 2. The absence rate ranged between 3.67 and 5.01%, with 4.19% of the workforce absent each month on average. Despite an expected decrease in absence rate based on historical data, the rate surged in March 2020, peaking at 6.19% in April 2020, the highest sickness absence rate recorded in a month since 2015, before decreasing to typical levels by July. The sickness absence rate rose towards the end of 2020 and peaked in January 2021 at 5.74%: a winter all-time high. It decreased towards a typical 3.97% in March 2021 and then continuously climbed upwards until the end of available data. From July 2021 until March 2022, the sickness absence rate for each month was consistently at least 1% higher than the previous year. A new record high of 6.68% was reached in the winter peak of January 2022. The typical seasonality in sickness absence rates (peaks in winter, troughs in summer) are driven by four categories of absence: cold/cough/flu, chest and respiratory problems, gastrointestinal problems, and anxiety/depression/other psychiatric illness, Fig A1 in Multimedia Appendix 1: Additional Figures.

Our expectation that the indicators of COVID-19 severity were positively correlated with the absence rate is hinted at by the surge in absences in March 2020 in Fig 2. Fig 3 shows that reasons for absence in our COVID-19-related category

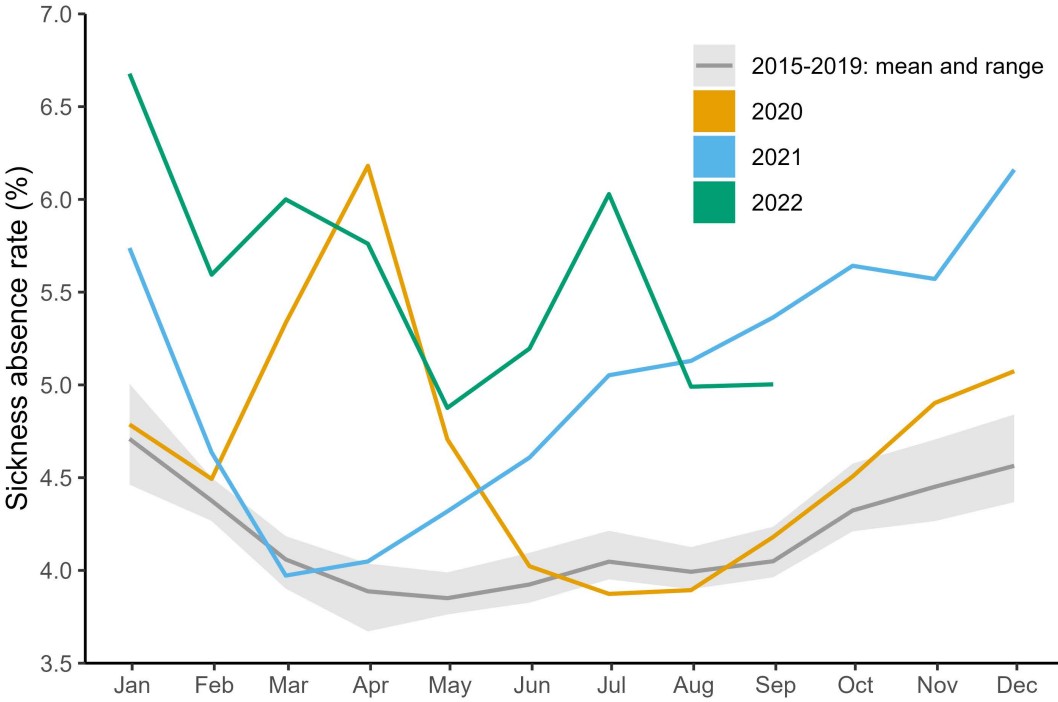

**Fig 2. Overall sickness absence rates in NHS England staff by month.** The yellow, blue and green lines indicate the overall sickness absence rates by month for 2020, 2021, and 2022 respectively. The dark grey line shows the mean sickness absence rate for a fixed month in 2015-2019, and the light grey region highlights the minimum and maximum rate during this period.

(S13 cold/cough/flu, S15 chest/respiratory, and S27 infectious disease) were responsible for the surge in sickness absence rates in March 2020. Absence rates for these reasons then returned to their previous sinusoidal cycle but were amplified in magnitude, explaining the winter 2020 peak seen in Fig 2.

S10 anxiety/depression/other psychiatric illness (mental health) was consistently the main reason for staff absence from January 2015 to September 2022. It accounted for 20% of the staff absences on average each month, Fig 3. Furthermore, sickness absence for mental health reasons has increased yearly since 2016, with bi-annual peaks in July and December, Fig 4(a). This trend appears to be slightly disrupted after March 2020, with the July peaks of 2020 and 2021 being high relative to the December ones, and a deep trough in April 2021.

The average sickness absence rate for our COVID-19-related category (S13 cold/cough/flu, S15 chest/respiratory, and S27 infectious disease) more than doubles when we compare 2015−2019–2020−2022 (0.49% vs 1.24%). The low of summer troughs and the duration and peak of winter waves of sickness absence rates were significantly higher from 2020 onwards, Fig 4(b). Furthermore, Fig 4(a) and (b) combined suggest that the huge rise in absences in mid-2021 (described earlier) was driven by a combination of COVID-19 related and mental health-related sickness absence.

### COVID-19-related absence: Linear regression models

Across the univariate linear regression models developed using data between July 2020 and December 2021, COVID-19 positivity according to the ONS data was the strongest predictor of absence (AIC = 3.5, adj. $R^2$ = 76.7%), Table 1. The number of new COVID-19 hospital admissions as a univariate predictor (AIC = 9.5, adj. $R^2$ = 67.6%) outperformed the number of positive PCR COVID-19 tests (AIC = 12.1, adj. $R^2$=62.3%), as well as the ONS estimated COVID-19 incidence (AIC = 16.813, adj. $R^2$=51.4%).

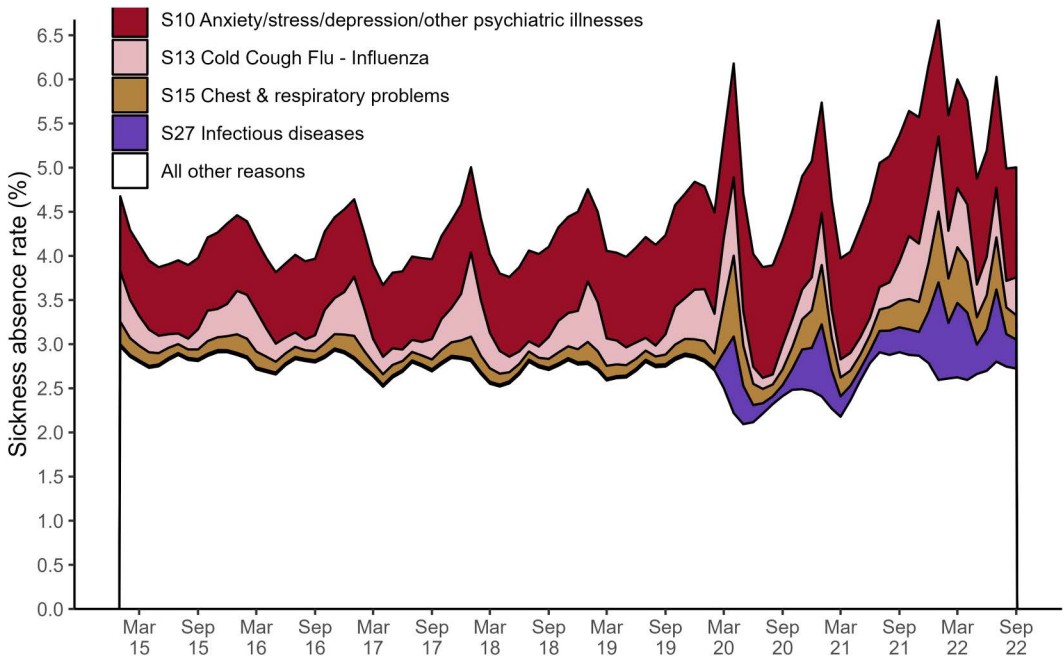

**Fig 3. Overall sickness absence rates in NHS England staff by month, broken down by reason.** The main reason behind a member of staff's sickness absence was recorded in the electronic staff record. This timeseries shows the monthly observations of sickness absence rates from January 2015 until the end of September 2022. Each colour indicates the proportion of the overall monthly sickness absence rate attributed to either S13 cold/cough/flu (pink), S15 chest and respiratory problems (dark yellow), S27 infectious disease (purple), S10 anxiety/stress/depression/other psychiatric illnesses (red), or any other reason (white). Multimedia Appendix 1: Additional Figures provides a more detailed version of this figure, giving a breakdown for all reasons (Fig A1).

The univariate model of hospitalisations explained the wave of absences in December 2020 – February 2021 well in magnitude and timing (Fig 5(b)), whereas univariate models of the other three predictors (incidence Fig 5(e), tests Fig 5(h), and positivity Fig 5(k)) underestimated this wave. However, the magnitude of the underestimation was much smaller for the ONS positivity predictor. These other three predictors (estimated incidence and average positivity according to the ONS, PCR positive tests) better estimate the wave of absences in late 2021 (post-June). The univariate model of hospitalisations underestimates absences during this time. The univariate models of tests, estimated incidence and average positivity according to the ONS estimate a similar absence rate trend, Fig 5(e,h,k), typically overestimating the July 2020 trough and rise in June 2021, while underestimating the peak in January 2021, but matching late 2021 until 2022 well.

The better performing multivariate regression models include the hospitalisation predictor, Table 1. The two predictor multivariate models combining hospitalisations with either ONS positivity ($F_{(2, 15)}$ = 132.63, $P < .001$) (AIC = −19.8, adj $R^2$ =93.9%), ONS incidence ($F_{(2, 15)}$ = 103.906, $P < .001$) (AIC = −15.691, adj $R^2$ =92.4%) or positive tests ($F_{(2, 15)}$ = 102.44, $P < .001$) (AIC = −15.5, adj $R^2$ =92.3%) generate a significant improvement in prediction error and adjusted $R^2$, compared to the best univariate model (ONS positivity; AIC = 3.5, adj. $R^2$ = 76.7%), Table 1. In contrast, the two variable multivariate models combining tests and ONS estimated positivity ($F_{(2, 15)}$ = 38.187, $P < .001$) (AIC = 0.354, adj $R^2$ = 81.4%), or combining ONS estimated incidence and average positivity ($F_{(2, 15)}$ = 43.854, $P < .001$) (AIC = −1.751, adj $R^2$ = 83.4%), provide only a slight improvement in predictive power (reduction in prediction error), Table 1. The multivariate model combining tests and ONS estimated incidence ($F_{(2, 15)}$ = 24.704, $P < .001$) (AIC = 6.649, adj $R^2$ = 73.6%) performs slightly worse than the best univariate model (ONS positivity; AIC = 3.5, adj. $R^2$ = 76.7%), Table 1. These suggest that the testing, ONS incidence and ONS average postivity data streams contain similar additional information related to paths of staff infection (and therefore absence) not captured by the hospitalisation data stream alone.

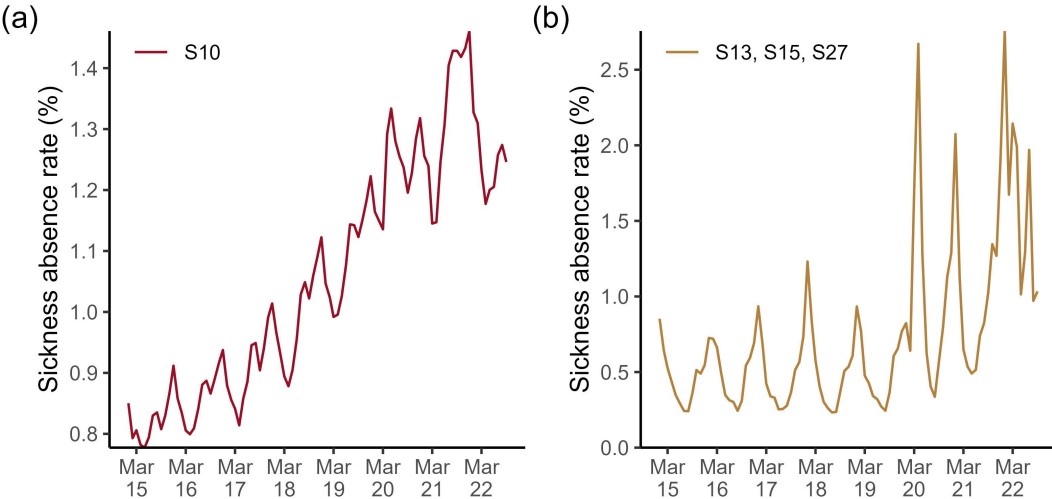

**Fig 4. Sickness absence rates in NHS England staff by month for a specific reason.** Each panel shows the trend over time for (a) absence in the mental health-related category; (b) absence for the COVID-19-related category (S13 cold/cough/flu, S15 chest and respiratory problems, or S27 infectious disease).

The multivariate model of ONS estimated positivity and hospitalisations was the best performing model overall (lowest AIC, highest adjusted $R^2$), Table 1. The model matches the trend in absences (June 2020-June 2021) well overall, Fig 6(a), fitting the timing and magnitude of the peaks and troughs around the winter 2020 wave of absences. However, the model estimated that between June and December 2021, there would be a rise, constant level, then a rise again in absence rates when instead there was a steady rise. This is evident in figure Fig 6(b), where some observed absence rate values between 0.6% and 1.5% are over- and under-estimated. We test the assumptions of this multivariate regression model in Multimedia Appendix 1: Assumption testing of best-performing multivariate regression model.

Neither the multivariate model of ONS and hospitalisations (Fig 6(a)) nor the univariate models (Fig 5(b,e,h,k)) were able to completely explain COVID-19 sickness absence rates between January and July of 2022. The multivariate model of ONS and hospitalisations matches the peak of absence in January 2022 (3%), Fig 6(a). Additionally, it correctly estimates that February rates would drop, but estimates a lower drop (3% vs 1.5%). However, the model estimated that COVID-19-related absence rates would be at an all-time high in March and still high in April, when instead, they were 1% lower – and lower than in January 2022. The model captures the fall from April into July, where the modelled and observed rates coincide again at 1%. Similarly, the univariate models struggle to explain the trend between February and May 2022, either vastly underestimating (see tests Fig 5(h), hospitalisations Fig 5(b)) or overestimating (incidence Fig 5(e), positivity Fig 5(k)) the absence rate during this period.

Our results indicate that the the ONS incidence and positivity estimates, as well as PCR positive test, contain similar information about absences. Firstly, the univariate models of tests, estimated incidence and average positivity according to the ONS estimate a similar absence trend, Fig 5(e,h,k), as discussed previously. Additionally, there is no improvement in multivariate model performance from combining two of these predictors in the same model, Table 1. Finally, by testing for multicollinearity in the multivariate regression models (Table 1) using the variance inflation factor (VIF) [25] we find evidence to support that these predictors contain similar information. The VIF was low in two predictor models combining hospitalisations with ONS incidence, ONS positivity or positive tests (between 1.2 and 1.4 for each predictor). In contrast, it was high for the multivariable model combining

**Table 1. Regression models estimating the COVID-19 related sickness absence rate.** Coefficients were estimated using data between July 2020 and December 2021. Each numbered row in the table indicates a different model. Columns 2-5 contain the regression coefficients (top) and their corresponding standard error (bottom), with the significance of the coefficient indicated by the number of asterisks. The following asterisk system is to indicate the significance of $P$ values: $P < .1$; * $P < .05$; ** $P < .01$; *** $P < .001$.

| Model | Independent Variable Coefficients (Standard Error) | | | | | Observations | Adjusted $R^2$ | Akaike Information Criterion (AIC) | F-statistic |
|---|---|---|---|---|---|---|---|---|---|
| | Incidence (ONS) | Positive Tests | Hosp. Admissions | Avg. Positivity (ONS) | Constant | | | | |
| (1) | $1.1 \times 10^{-4}$*** | | | | 6.41*** | 18 | 0.514 | 16.813 | 18.953*** |
| | $(2.52 \times 10^{-5})$ | | | | (0.108) | | | | ($df = 1$; 16) |
| (2) | | $5.93 \times 10^{-7}$*** | | | 0.566*** | 18 | 0.626 | 12.090 | 29.440*** |
| | | $(1.09 \times 10^{-7})$ | | | 0.1 | | | | ($df = 1$; 16) |
| (3) | | | $1.68 \times 10^{-5}$*** | | 0.541*** | 18 | 0.676 | 9.523 | 36.404*** |
| | | | $(2.78 \times 10^{-6})$ | | (0.0947) | | | | ($df = 1$; 16) |
| (4) | | | | $9.33 \times 10^{-7}$*** | 0.443*** | 18 | 0.767 | 3.539 | 57.070*** |
| | | | | $(1.23 \times 10^{-7})$ | (0.0874) | | | | ($df = 1$; 16) |
| (5) | | $-7.4 \times 10^{-7}$* | | $1.96 \times 10^{-6}$*** | 0.36*** | 18 | 0.814 | 0.354 | 38.187*** |
| | | $(3.31 \times 10^{-7})$ | | $(4.74 \times 10^{-7})$ | (0.0864) | | | | ($df = 2$; 15) |
| (6) | $-3.1 \times 10^{-4}$* | $2.1 \times 10^{-6}$** | | | 0.46*** | 18 | 0.736 | 6.649 | 24.704*** |
| | $(1.12 \times 10^{-4})$ | $(5.53 \times 10^{-7})$ | | | (0.0925) | | | | ($df = 2$; 15) |
| (7) | $-1.17 \times 10^{-4}$* | | | $1.71 \times 10^{-6}$*** | 0.348*** | 18 | 0.834 | −1.751 | 43.854*** |
| | $(4.27 \times 10^{-5})$ | | | $(3.02 \times 10^{-7})$ | (0.0814) | | | | ($df = 2$; 15) |
| (8) | $7.72 \times 10^{-5}$*** | | $1.33 \times 10^{-5}$*** | | 0.41*** | 18 | 0.924 | −15.691 | 103.906*** |
| | $(1.06 \times 10^{-5})$ | | $(1.43 \times 10^{-6})$ | | (0.0493) | | | | ($df = 2$; 15) |
| (9) | | $4.07 \times 10^{-7}$*** | $1.19 \times 10^{-5}$*** | | 0.401*** | 18 | 0.923 | −15.451 | 102.435*** |
| | | $(5.54 \times 10^{-8})$ | $(1.51 \times 10^{-6})$ | | (0.0501) | | | | ($df = 2$; 15) |
| (10) | | | $9.89 \times 10^{-6}$*** | $6.41 \times 10^{-7}$*** | 0.361*** | 18 | 0.939 | −19.820 | 132.632*** |
| | | | $(1.45 \times 10^{-6})$ | $(7.63 \times 10^{-8})$ | (0.0462) | | | | ($df = 2$; 15) |
| (11) | $4.79 \times 10^{-5}$ | $1.53 \times 10^{-7}$ | $1.28 \times 10^{-5}$*** | | 0.406*** | 18 | 0.919 | −13.839 | 65.225*** |
| | $(8.68 \times 10^{-5})$ | $(4.51 \times 10^{-7})$ | $(2.16 \times 10^{-6})$ | | (0.0521) | | | | ($df = 3$; 14) |
| (12) | $8.88 \times 10^{-6}$ | | $1.02 \times 10^{-5}$*** | $5.72 \times 10^{-6}$ | 0.365*** | 18 | 0.935 | −17.894 | 82.887*** |
| | $(3.69 \times 10^{-5})$ | | $(2.07 \times 10^{-6})$ | $(2.98 \times 10^{-7})$ | (0.0510) | | | | ($df = 3$; 14) |
| (13) | $-1.46 \times 10^{-4}$ | $2.31 \times 10^{-7}$ | | $1.58 \times 10^{-6}$* | 0.351*** | 18 | 0.824 | 0.139 | 27.483*** |
| | $(1.08 \times 10^{-4})$ | $(7.86 \times 10^{-7})$ | | $(5.43 \times 10^{-7})$ | (0.0844) | | | | ($df = 3$; 14) |
| (14) | | $-5.04 \times 10^{-8}$ | $9.68 \times 10^{-6}$*** | $7.17 \times 10^{-7}$ | 0.357*** | 18 | 0.935 | −17.880 | 82.818*** |
| | | $(2.33 \times 10^{-7})$ | $(1.79 \times 10^{-6})$ | $(3.63 \times 10^{-7})$ | (0.0510) | | | | ($df = 3$; 14) |
| (15) | $7.18 \times 10^{-5}$ | $-4.51 \times 10^{-7}$ | $1.08 \times 10^{-5}$*** | $7.66 \times 10^{-7}$ | 0.362*** | 18 | 0.934 | −16.992 | 61.558*** |
| | $(7.89 \times 10^{-5})$ | $(4.99 \times 10^{-7})$ | $(2.17 \times 10^{-6})$ | $(3.69 \times 10^{-7})$ | (0.0515) | | | | ($df = 4$; 13) |

ONS incidence and positive tests ($\approx 36$), ONS incidence and ONS average positivity ($\approx 8.5$), or ONS positivity and positive tests ($\approx 18$). Additionally, in the three or four predictor models, the VIF for hospitalisations remained < 3. However, for any other predictors (positive tests, ONS incidence, or ONS average positivity) the VIF was > 15. This suggests that there is little correlation between hospitalisations and positive tests, ONS incidence, or ONS average positivity but that there are high correlations between the latter three.

The scatterplots of predictor against fitted values with the univariate regression line, (Fig 5(a,d,g,i)) contain outliers at very large values of the predictor. For example, there are two outliers in the plot for the hospitalisations univariate model Fig 5(a). One where the hospitalisation rate is > 100,000 and the absence rate is 2%, and another where hospitalisation is closer to 35,000, and the absence rate is 2%. These outliers suggest a problem

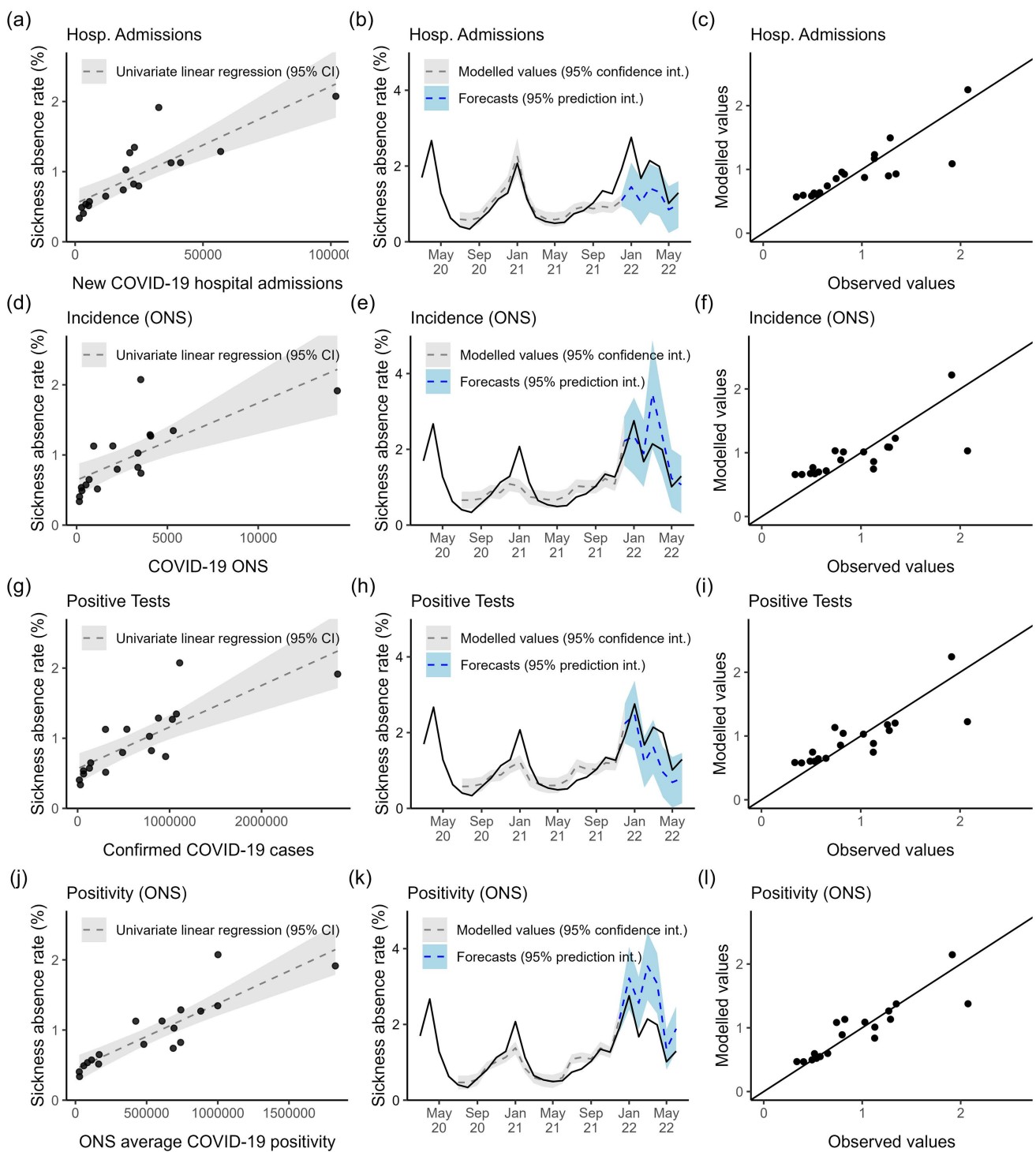

**Fig 5. Univariate models of COVID-19-related absence.** The panels in each row contain results for a different univariate predictor; new COVID-19 hospitalisations (a-c), ONS estimated COVID-19 incidence (d-f), confirmed PCR-positive COVID-19 tests (g-i), ONS estimated average COVID-19 positivity (j-l). The panels in each column show a different visualisation for a univariate model. (a,d,g,j) Scatterplots of predictor against the absence rate, including the fitted regression line. (b,e,h,k) Timeseries of modelled values (dashed grey) with 95% confidence interval (grey, shaded), forecasts (dashed blue) with 95% prediction interval (blue, shaded), and observed sickness absence trend (solid black). (c,f,i,l) Scatter plot of modelled values against the observed values. Black line is the theoretical line of equality (modelled = observed).

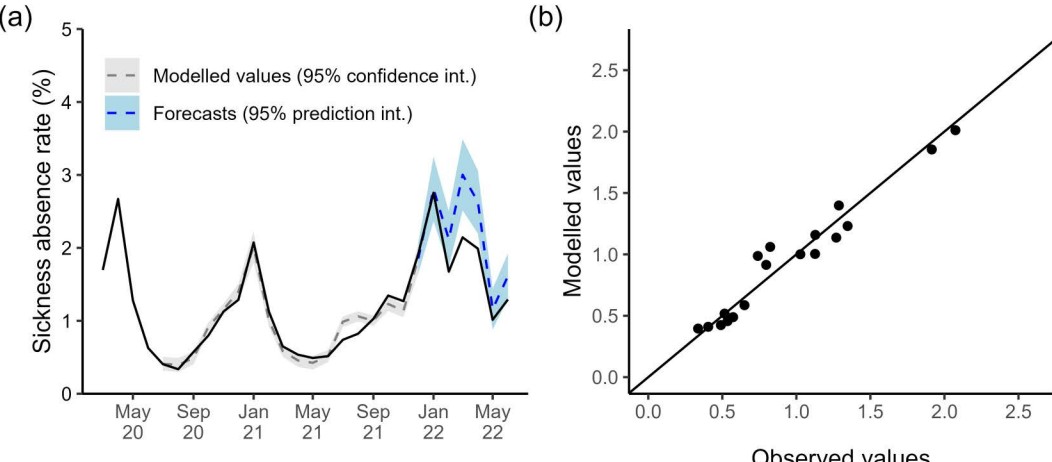

**Fig 6. COVID-19 related absence as a multivariate regression model of new hospitalisations and ONS estimated COVID-19 positivity.** (a) Timeseries of modelled values for 2020–2022 (dashed grey) with 95% confidence interval (grey, shaded), predictions for 2022 (dashed blue) with 95% prediction interval (blue, shaded), and observed sickness absence trend (solid black). (c) Scatter plot of modelled values against the observed values. Black line is theoretical line of equality (modelled = observed).

with scale and could be skewing the linear relationship, causing the poorer fit to absences in late 2021. Alternatively, it may suggest a threshold in hospitalisations where further increase does not impact the absence rate. Some evidence of heteroscedasticity in Fig 5(a,d,g,i) suggests right skewness of the surveillance data (i.e., the predictors). However, log transformations of the surveillance data did not improve the model fits (results not shown).

## Forecasting mental health-related absence

A SARIMA(0, 1, 3)(0, 1, 0) [12] model fit the mental health-related absence rates from January 2015 until March 2020 best ($AIC = -238.46$), Fig 7(a). This model has a moving average (MA) part of order 3 with a 1st-order difference and a 12-month period. The next month's absence rate is a linear combination of the previous month, the same month in the previous year, and the month prior from the previous year, plus a new white noise term and the last three months' noise terms.

The SARIMA model trained on pre-COVID-19 data did not pick up the spike in mental health-related absences in April and May 2020 (which coincide with the first wave of COVID-19), instead estimating a slower rise in absences that would peak in July 2020, typical of previous years. Excluding these two early months, the model explains the increasing trend with seasonal peaks in from July 2020 until November 2021 well, with estimates falling within 95% intervals, Fig 7(b). However, the model overestimates the January 2022 peak and estimates a typical rise in absence rates in March/April 2022 when, instead, there was a deeper drop than usual after the winter peak, Fig 7(a). The model continues to significantly overestimate absence rates through the summer of 2022 until the last data point in September.

We trained a time series model to include the months between March 2019 – December 2021, and a SARIMA(2, 0, 0)(0, 1, 1) [12] with drift fit best with $AIC = -307.85$. This model has non-seasonal autoregressive (AR) part order 2 with no differencing, and a seasonal part with AR order 1, MA order 1 and a 1st and 12th order difference. The next month's absence rate is a linear combination of the last two previous months, the same month in the previous year, and the two

months prior from the previous year, plus a new white noise term and the noise terms for that month in the previous year.

The observed mental health sickness absence trend for January-March 2022 falls within this model's 95% prediction interval, suggesting an initial good fit. The model captures the decreasing trend but slightly overestimates absences for each month, Fig 7. However, the model also estimates a rise in absence rates in March/April 2022 when instead, there was a deeper drop than usual after the winter peak, Fig 7(a). The model continues to overestimate absence rates through the summer of 2022 until the last data point in September, with May-September falling outside the 95% prediction interval.

## Discussion

This study highlights the impact of COVID-19 on sickness absence rates in NHS England from late 2020 into 2022. Our first aim was to identify the key trends and sources of variability in NHS England's Sickness absence rates during this period. The second was to investigate whether we can use this information to develop models that explain the sickness absence rates. Consequently, we developed regression models with COVID-19 activity indicators as predictors to understand the relationship between COVID-19 and NHS staff absences in the COVID-19-related category (S13, 15, 27). Additionally, to examine the impact of COVID-19 on mental health-related sickness absence (S10), we developed deterministic time series models that extrapolated pre-March 2020 data.

The data revealed a notable increase in sickness absence rates starting around March 2020, which coincided with the establishment of COVID-19 in England. Additionally, we observed another surge in absence during late 2020, and since mid-2021, the levels have remained consistently high. Consequently, except for a few months in 2020 and 2021, the overall sickness absence rate has progressively risen since March 2020. As a result, each month's absence rates are at least 1% higher in 2022 compared to pre-pandemic levels. The primary contributors to this variation in sickness absence rates, notably during the surge around March 2020, were the mental health and COVID-19-related categories.

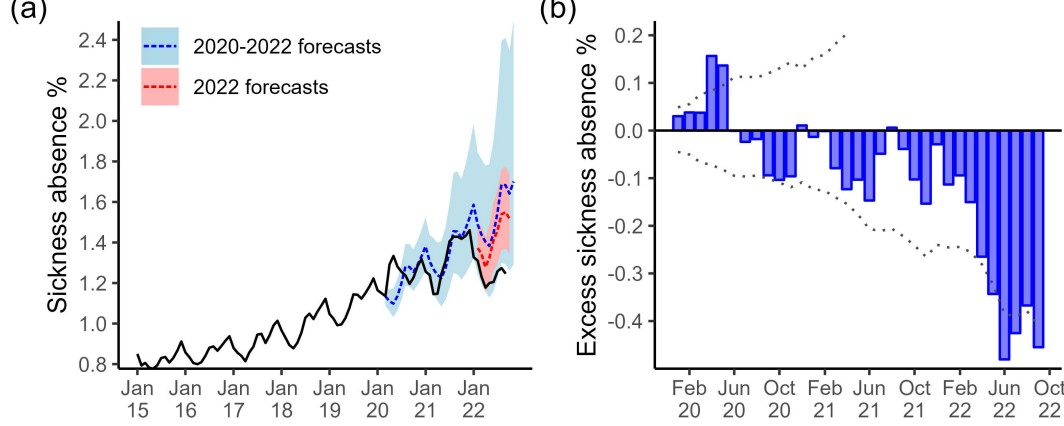

**Fig 7. SARIMA models of mental health-related sickness absence.** (a) Time series showing the observed trend (solid black line) and predictions. We used a SARIMA model trained with 2015–2019 to predict the 2020–2021 trend (dashed blue line, with the light blue shaded region indicating prediction interval). Another model was trained with 2015–2021 data to predict the 2022 rates (dashed red line, with the light red shaded region indicating prediction interval). (b) Barplot showing the difference between the observed trend and the predictions from the SARIMA model trained on 2015–2019 data (shaded blue bars). Bars above the horizontal zero line (solid black) are months where the observed trend was higher than the model predictions. The dotted line indicates the difference between the observed trend and the 95% prediction interval.

The data showed that the average rate of mental health-related absences increased consistently year on year between 2017 and 2022. However, our modelling results indicate that there was a spike in mental health-related absence in early 2020, and suggest that the long-term increase may have been slowing in 2022. COVID-19 may have caused a sudden surge or shock in mental health-related sickness absence among NHS staff during the first wave of the pandemic (e.g., possibly stress/anxiety or burnout related). We observed a large deviation from the expected trend according to our time series models, which extrapolated pre-March 2020 data, in April and May 2020. Apart from this period, the deterministic model, characterised by an increasing trend with bi-annual peaks, effectively accounted for the overall trend between June 2020 and November 2021. In addition, Van der Plaat et al. [12] demonstrated a regional correlation between the increase in mental health sickness absence in March-April 2020, compared to 2019, and the cumulative prevalence of COVID-19-related sickness absence during the same period. The authors hypothesized that the spike in mental health-related sickness absence was driven by the combined stresses experienced by healthcare workers, both at work and in their personal lives, as a consequence of the epidemic.

The time series model overestimates the mental health-related absence rate between January and September 2022, which suggests a shift in the underlying dynamics. A possible explanation is that the continuous rise in mental health absences from 2016 was beginning to plateau by 2020. Then, the arrival and constant pressure from COVID-19 had knock-on effects on the well-being of staff and caused a significant increase in mental health absences, but these began to alleviate in 2022. Alternatively, the underestimation could be explained by another reason, such as changes in NHS internal policy regarding absences in this category.

Indicators of COVID-19 activity correlated positively and strongly with absence in our COVID-19-related category (S13, 15, 27) over the study period, but the relationship changed dynamically over time, aligning with the concept of temporal stratified heterogeneity. As we will discuss in the following paragraphs, different predictors best explain different waves of absence, particularly as new variants emerged and vaccines were introduced. The best-fitting multivariate model could explain most of the variability in absence rates in these categories, suggesting that COVID-19 was the main driver behind them. Additionally, our regression models suggest two independent sources of infection risk for NHS staff: community transmission and hospital exposure.

The estimated COVID-19 positivity, according to the ONS, was the strongest univariate predictor of COVID-19 related sickness absence over the July 2020 to December 2021 time period, followed by the number of new COVID-19 hospitalisations, then the number of PCR-positive tests, followed by the ONS incidence rate. This suggests that the ONS positivity data stream contained the most information about how NHS staff caught COVID-19 infections (becoming ill and absent) and that this source is essential to estimate absences. This result supports Zheng et al. [26], who showed that COVID-19 rates in NHS staff mainly rose and declined in parallel with the number of community cases. The ONS data reflects the COVID-19 positivity of people in private house-holds, which includes the households of NHS staff [20]. Therefore, it is intuitive that it correlates so strongly with COVID-19 sickness absence. However, as a univariate model of this data stream alone does not best explain the staff absence rate, which is essentially a proxy for staff infections, this supports previous findings that staff face more significant infection pressure than a random member of the general population (living in private households) [5,20]. Furthermore, the reported PCR-positive tests/cases include people who are (or will end up) hospitalised [15]. They are likely not the strongest predictor between March 2020 and December 2021 because the policy around groups tested and the testing capacity changed over this period [27]. Tests may add slightly different information than the ONS estimated positivity to estimate sickness absence since they were widely reported and could influence behaviour.

Our results suggest that the ONS estimated positivity, ONS incidence, and the positive test data streams contain similar information about COVID-19-related sickness absence in NHS staff. First, the univariate models of tests, ONS incidence or positivity estimate a similar absence trend. Second, the multivariate models combining hospitalisations with the testing

or either ONS stream perform similarly, and including the third predictor does not add to the performance. Furthermore, the variance inflation factor was severely high for these predictors when included in the same model, suggesting a strong correlation. We hypothesise that the tests and ONS predictors provide information about absences due to infection pressure in the community.

The number of new COVID-19 hospitalisations was the second strongest univariate predictor of COVID-19-related sickness absence over the July 2020 and December 2021 period. COVID-19 hospital admissions are a direct source of infection for NHS staff since they primarily work in hospitals and will be exposed to nosocomial COVID-19 infections there. Furthermore, a surge in hospitalisations should correlate with (can be driven by) a large number of infections in the community, another source of infection. This predictor may indirectly capture this path. In previous work using multivariate regression to link influenza activity to sickness absence, Schanzer et al. [16] used laboratory-confirmed H1N1/2009 hospital admissions as a proxy for influenza activity instead of the number of laboratory-confirmed cases overall. Although the scale of COVID-19 has far exceeded seasonal influenza, there is a similarity there.

We found that a univariate regression model of new COVID-19 hospital admissions better estimated the December 2020 – February 2021 wave compared to the other predictors. However, the positive tests or ONS-estimated infections or positivity better estimated the September 2021 – December 2021 wave. One possible contributing factor to this dynamic is the rollout of COVID-19 vaccines (it began in December 2020, and by March and September 2021 $\approx 30$% and $\approx 70$% of adults had their first dose [28], respectively) reduced the number of people hospitalised (or the number of severe illnesses) relative to the number infected [29]. The rise in milder but more transmissible sub-variants of COVID-19 may have had a similar effect [30,31]. The Alpha variant was most common between 8th December 2020–17th May 2021. Subsequently, the slightly more transmissible but equally severe Delta variant was most common from 17th May to 19th December 2021, and the far more transmissible but less severe Omicron variant dominated from 20th December 2021 onwards [32]. Furthermore, suppose we assume admissions are a proxy for the infection pressure in the workplace for most NHS staff (hospitals). Vaccines reducing the number of admissions reduces the infection pressure from this source and may explain why hospitalisation is a stronger predictor for the December 2020 – February 2021 wave. To conclude, each information stream (i.e., ONS positivity, incidence or positive tests vs admissions) explaining different periods during March 2020-December 2021 also i) demonstrates why they perform well when combined in a multivariate model and ii) highlights that the relationship between COVID-19 absence and surveillance data was dynamic over the study period.

The regression model explaining COVID-19 sickness absence between July 2020 and December 2021 best combines these two streams (ONS estimated positivity, new admissions) into a multivariate model and can explain most of the variability in COVID-19-related sickness absence rates between March 2020 and December 2021. Compared to the univariate ones, the strong performance of this model suggests that each stream contains independent information about the sources of infection for NHS staff (pathways NHS staff can catch infection). These are infection pressure from the community and additional infection pressure from working in hospitals.

The multivariate model combining admission and ONS estimated positivity estimates COVID-19-related sickness absence between July 2020 and early 2022 well before significantly overestimating March-May. This suggests that other dynamics emerged that a statistical model with constant parameters could not capture, for example, the rise in milder but more transmissible sub-variants of COVID-19 in late 2021 [32], as well as changes in policy or the perception of COVID-19 risk. Furthermore, the regression coefficients hide changes in the dynamics. This is evident in the univariate model of hospitalisations explaining the trend between July 2020 and 2021 well, but not afterwards. Another factor that may have weakened the regression models' performance into late 2021 is Long Covid, also known as post-acute COVID-19 syndrome [33]. A subset of individuals

infected with COVID-19 experience these prolonged symptoms, including fatigue, breathlessness, and cognitive impairment, which can impact their ability to work [34,35]. ONS data estimates that approximately 1 in 10 individuals infected with COVID-19 in the UK experienced symptoms persisting for 12 weeks [36], with 2.8% of the UK population self-reporting symptoms in April 2022 [37]. These individuals may not have been captured by the COVID-19 surveillance data streams while they remained absent from work. Long Covid may also help explain the overall increased sickness absence rates since 2020. Cases may have been recorded under our COVID-19-related sickness absence category (which includes S13 Cold, Cough, Flu; S15 Chest & Respiratory Problems; and S27 Infectious Diseases). Alternatively, since Long Covid has been linked to persistent psychological and neurological effects [38–40], it may have contributed to the increases in mental health-related sickness absence.

Our multivariate model could estimate the impact of future waves of COVID-19 on staff absence rates. For example, we could use predictions of hospitalisations and incidence (with 95% confidence intervals) to create short-term projections for staff absence rates. However, given that our results suggest changes in how the predictors correlate with COVID-19 sickness absence rates, it could be more appropriate to train models with a bias towards more recent surveillance data or use dynamic regression models. The most recent surveillance data reflect the current picture regarding COVID-19 variants, vaccination, non-pharmaceutical interventions, and policy. In contrast, regression models trained over longer periods hide the recent dynamics in their coefficients.

A further limitation of our linear regression models is that they do not explain the dynamics behind COVID-19-related absences. They do not give us a way to understand the relationship between hospitalisations and the community on absences other than highlighting that these two sources play a role. We explored transforming the predictor variables (using natural log and quadratic power function – not shown), but this did not improve the performance of the models. Therefore, it is unclear what the exact relationship is between COVID-19 absences and the proxy variables for COVID-19, outside of them being strong and positive. Furthermore, there are variations in local epidemiology and different categories of staff, which our approach does not consider. For example, sickness absence rates are consistently higher in ambulance staff [14] and lower in non-patient-facing roles [19]. We also do not consider COVID-19 vaccination, which reduces the likelihood of NHS staff and their families falling ill with COVID-19 [10]. Our framework does not include other seasonal respiratory infections, such as influenza and rhinovirus, which will contribute to absence in the S13, 15, and 27 categories (our COVID-19-related category).

The data used in our study was collected for the whole of England, with no additional spatial or socioeconomic information. However, both socioeconomic deprivation [41,42] and the degree of urbanicity (e.g., urban vs. rural differences) [43] have been linked to increased risk of contracting COVID-19 and experiencing severe outcomes in England. Additionally, socioeconomic deprivation has previously been identified as an indicator of increased sickness absence in a large NHS organization [44]. If such data were accessible, future research could explore how these spatial and socioeconomic factors influence the relationship between COVID-19 and sickness absence in HCWs across different regions and population groups.

Our study shows an increasing trend of mental health-related sickness absence among NHS healthcare workers, highlighting the importance of continued investment in mental health support for workers. However, our time series analysis approach is unable to explicitly identify the driving factors behind this long-term trend. Additionally, we have not explicitly linked indicators of COVID-19 activity to mental health-related absence. How to define the relationship and incorporate a shock function to reflect the pressure on NHS staff is unclear. However, a first step for future investigation could involve incorporating COVID-19 activity indicators as regressors in a time series model, thus providing further insight into the relationship between COVID-19 and mental health-related absences. Moreover, identifying the driving factors behind the long-term mental health-related absences using individual worker-level covariates is a key area for future research.

## Multimedia Appendix 1: Additional Figures

See Fig A1.

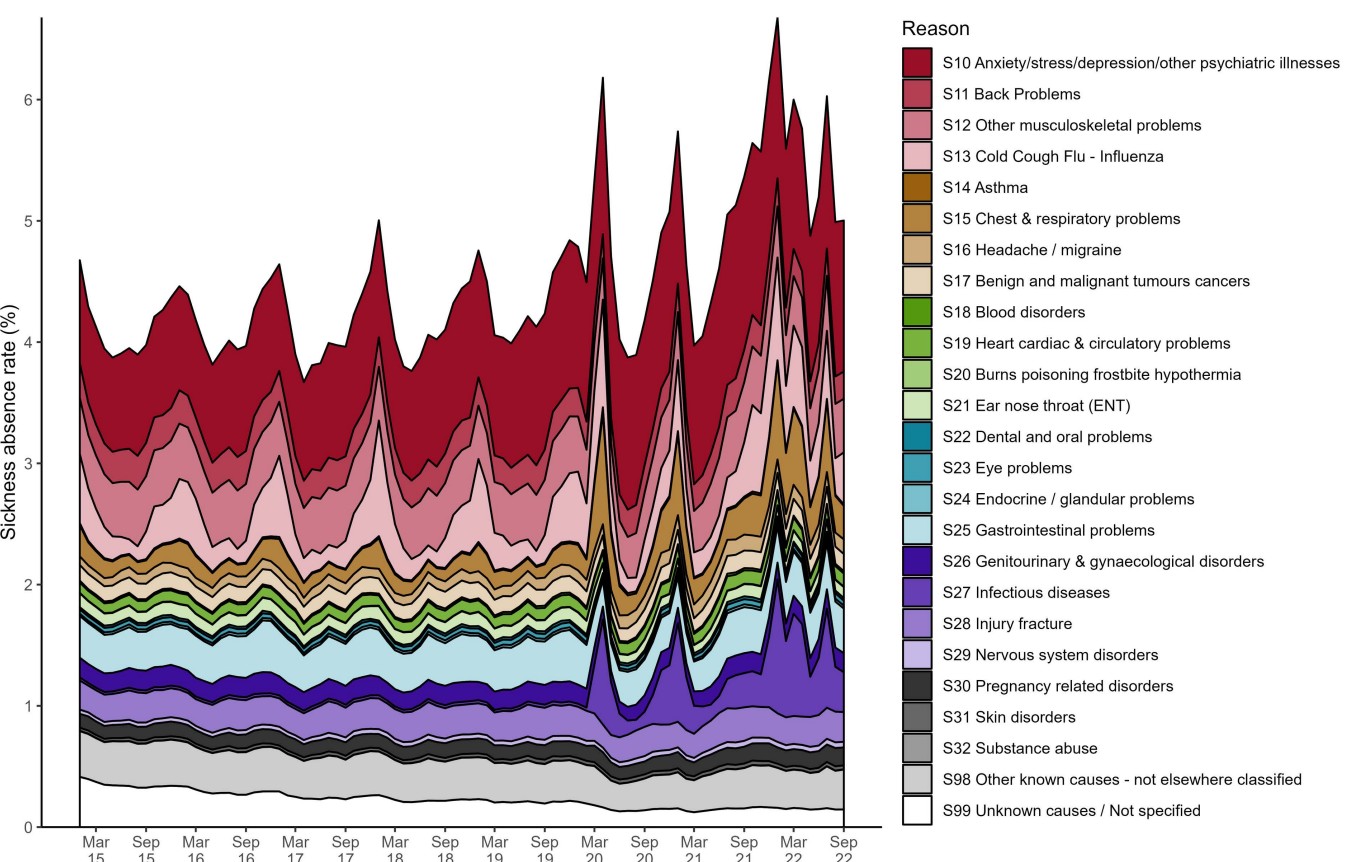

**Fig A1. Overall sickness absence rates in NHS England staff by month, broken down by reason.** The main reason behind a member of staff's sickness absence was recorded in the electronic staff record. This timeseries shows the monthly observations of sickness absence rates from January 2015 until the end of September 2022. Each colour indicates the proportion of the overall monthly sickness absence rate attributed to each main reason for absence.

## Multimedia Appendix 1: SARIMA model equations

### Fitted SARIMA models used to forecast mental-health-related absence

A SARIMA(0, 1, 3)(0, 1, 0) [12] model fit the mental health-related absence rates from January 2015 until March 2020 best (minimising AIC), Fig 7(a). The $\text{AIC} = -238.46$ and model equations are shown below

$$(1 - B)\left(X_t - X_{t-12}\right) = \left(1 - \theta_1 B - \theta_2 B^2 - \theta_3 B^2\right)\varepsilon_t.$$

Where $\theta_1 = -0.378$, $\theta_2 = 0.489$, $\theta_3 = -0.773$. The backshift operator ($B$) is defined as $B^k X_t = X_{t-k}$, where $X_t$ is the time series at time $t$, and $k$ is the number of time periods to shift the series backwards. This model has a moving average (MA) part of order 3 with a 1st-order difference and a 12-month period. The next month's absence rate is a linear combination of the previous month, the same month in the previous year, and the month prior from the previous year, plus a new white noise term and the last three months' noise terms.

Additionally, we trained a time series model to include the months between March 2020 – December 2021, as well as those between January 2015 and March 2020, and a SARIMA(2, 0, 0)(0, 1, 1) [12] with drift fit best with $AIC = -307.85$. The equation is shown below

$$\left(1 - \phi_1 B - \phi_2 B^2\right)\left(X_t - X_{t-12}\right) = \left(1 - \theta_1 B^{12}\right)\varepsilon_t + c_1.$$

Where $\phi_1 = 1.046$, $\phi_1 = -0.301$, $\theta_1 = -0.5271$, and $c_1 = 0.0065$. This model has non-seasonal autoregressive (AR) part order 2 with no differencing, and a seasonal part with AR order 1, MA order 1 and a 1st and 12th order difference. The next month's absence rate is a linear combination of the last two previous months, the same month in the previous year, and the two months prior from the previous year, plus a new white noise term and the noise terms for that month in the previous year.

## Multimedia Appendix 1: Assumption testing of best-performing multivariate regression model

We now test the assumptions of the best-performing multivariate regression model, which combines hospitalisations and ONS average positivity. The main assumptions of multivariate linear regression are as follows [23]. The relationship between the dependent variable and the independent variables is linear. The residuals (the differences between the observed and predicted values) follow a normal distribution, and their variance is constant across all levels of the independent variables (homoscedasticity). In other words, the spread of the residuals should remain roughly constant as the values of the independent variables change. Additionally, there should be no multicollinearity, there should not be a perfectly linear relationship between the independent variables. Finally, the residuals are independent of each other.

Figure A2(a) and Figure A2(b) show the relationship between each predictor and the residuals in the multivariate regression model. Each plot shows an approximately random scatter of points around the horizontal line at zero, which suggests that the assumption of linearity (between response and each predictor) is not violated. Additionally, the similar random scatter in the plot of residuals vs fitted values Figure A2 (c) with no clear trend suggests the linearity assumption is not violated. A Shapiro-Wilk test indicates there is evidence at the 5% significance level that the residuals are not normally distributed ($P$=0.02). Additionally, there is some evidence of an S shape in the QQ-normal plot, Figure A2 (d), which suggests a departure from normality (more extreme values than a normal distribution). A Goldfeld-Quandt test suggests no evidence of heteroscedasticity ($P = 0.097$) at the 5% significance level, i.e., that the residuals have constant variance. However, there are some slight increases in variability in the residuals for fitted values in the range of 1–1.4 in the plot of residuals vs fitted values, Figure A2 (a), relative to other fitted values. By calculating the variance inflation factor (VIF) (discussed below), we found no evidence of multicollinearity in the model combining ONS average positivity and hospitalisations. Since the response and predictors in the regression model are time series, the assumption of independence of residuals may likely be violated. Observations at adjacent time points are likely to be correlated. The Durbin-Watson test suggests evidence of autocorrelation at the 5% significance level ($P = 0.026$). The test statistic was 1.17, which falls outside of the 1.5–2.5 acceptable range and indicates some positive autocorrelation. The Autocorrelation Function (ACF) plot of the residuals, Figure A2 (f), shows a significant spike at lag 3, indicating a potential autocorrelation at that lag.

## Acknowledgments

We thank Dr. Colin Tilley and Dr. Morag MacPherson for their assistance in guiding and refining this paper. We would also like to acknowledge that John Bowers from the University of Stirling, a co-author of this work, is now deceased. EM accepts responsibility for the integrity and validity of the data collected and analyzed.

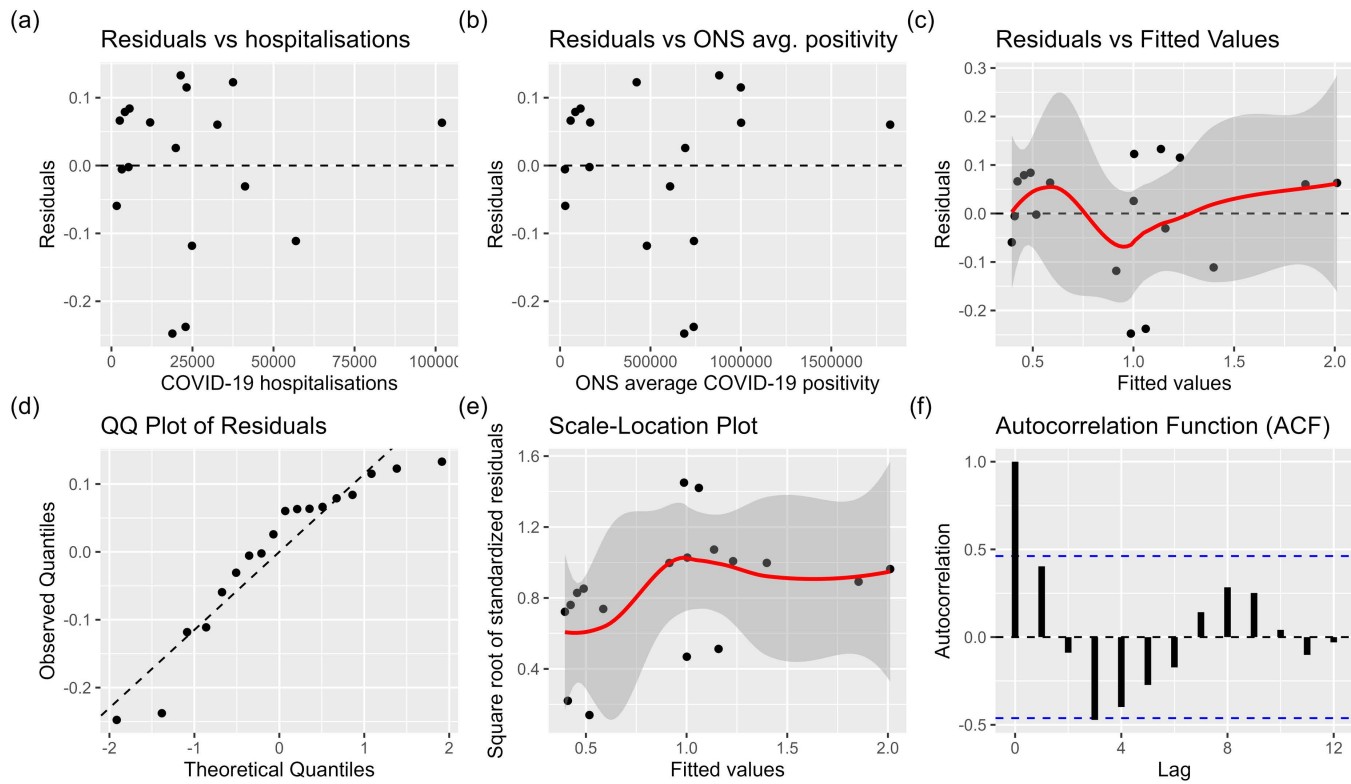

**Fig A2. Diagnostic plots to test assumptions of the multivariate linear regression model of new hospitalisations and ONS estimated COVID-19 positivity.** (a) Residuals vs fitted values; (b) Normal Quantile-Quantile (Q-Q) plot; (c) Scale-Location Plot; (d) Plot of residuals against the COVID-19 hospitalisations predictor; (e) Plot of residuals against the ONS average COVID-19 positivity predictor.

## Author contributions

**Conceptualization:** Ewan McTaggart, Itamar Megiddo, John Bowers, Adam Kleczkowski.

**Data curation:** Ewan McTaggart.

**Formal analysis:** Ewan McTaggart.

**Funding acquisition:** Adam Kleczkowski.

**Investigation:** Ewan McTaggart.

**Methodology:** Ewan McTaggart, Itamar Megiddo, John Bowers.

**Project administration:** Itamar Megiddo, Adam Kleczkowski.

**Resources:** Ewan McTaggart, Adam Kleczkowski.

**Software:** Ewan McTaggart.

**Supervision:** Itamar Megiddo, Adam Kleczkowski.

**Validation:** Ewan McTaggart.

**Visualization:** Ewan McTaggart.

**Writing – original draft:** Ewan McTaggart.

**Writing – review & editing:** Ewan McTaggart, Itamar Megiddo, Adam Kleczkowski.

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
