## [Decision Letter · Decision Letter 0]

Dear Dr. McTaggart,

Thank you for submitting your manuscript to PLOS ONE. After careful consideration, we feel that it has merit but does not fully meet PLOS ONE’s publication criteria as it currently stands. Therefore, we invite you to submit a revised version of the manuscript that addresses the points raised during the review process.

We look forward to receiving your revised manuscript.

Kind regards,

Md. Kamrujjaman, Ph.D

Academic Editor

PLOS ONE

Journal Requirements:

2. Please ensure that you include a title page within your main document. You should list all authors and all affiliations as per our author instructions and clearly indicate the corresponding author.

Reviewers' comments:

Reviewer's Responses to Questions

**Comments to the Author**

1. Is the manuscript technically sound, and do the data support the conclusions?

Reviewer #1: Yes

Reviewer #2: Yes

2. Has the statistical analysis been performed appropriately and rigorously?

Reviewer #1: Yes

Reviewer #2: No

3. Have the authors made all data underlying the findings in their manuscript fully available?

Reviewer #1: Yes

Reviewer #2: Yes

4. Is the manuscript presented in an intelligible fashion and written in standard English?

Reviewer #1: Yes

Reviewer #2: Yes

Reviewer #1: I would like to congratulate you for the effort in writing this article, from my point of view this article can be published with some changes, I suggest you just check the information in the paragraph they say:

“Another factor that may have weakened the regression models' performance into late 2021 is Long Covid, also known as post-acute COVID-19 syndrome [33]. A subset of individuals infected with COVID-19 experience these prolonged symptoms, including fatigue, breathlessness, and cognitive impairment, which can impact their ability to work [34,35]. These individuals may not have been captured…”

From my point of view the intervening factor may be long Covid, but it may also be many other factors, for this reason I suggest that if you do not have sufficient evidence that long Covid can produce this situation you can include in this paragraph long Covid and other factors, or if you think that long Covid is an important factor you should include information on the prevalence of long Covid in England at least as a justification for your comment.

Reviewer #2: This research can be recommended for publishing in this journal after revising the MS addressing these issues:

1. I have learned that the author used the SARIMA model. But it is not mentioned what type and how many parameters you considered for the mental health-related absence?

2. This study demonstrated that the mental-related absence is significant than S13, S15, and S27. Is this study able to find the key factors that increase the number of absences for mental health issues? At the same time, what is a suggestion for preventing mental health-related absences?

3. Why is this S10 (mental health-related absence) exponentially increased from March 17, whereas COVID-19 started in December 2019, as shown in Figure 2, panel (a)?

4. What is the limitation of this work and the scope of future study, which is not mentioned here?

**Do you want your identity to be public for this peer review?** For information about this choice, including consent withdrawal, please see our Privacy Policy

Reviewer #1: **Yes: ** DANTE ROGER CULQUI LEVANO

Reviewer #2: No

---

## [Author Response · Author response to Decision Letter 1]

9 Jun 2025

Response to Reviewers

We thank the reviewers for their thoughtful and constructive feedback. Below, we respond to each comment in turn, indicating how the manuscript has been revised or why specific changes were not made.

Reviewer #1

I would like to congratulate you for the effort in writing this article, from my point of view this article can be published with some changes, I suggest you just check the information in the paragraph they say:

“Another factor that may have weakened the regression models' performance into late 2021 is Long Covid, also known as post-acute COVID-19 syndrome [33]... These individuals may not have been captured…”

From my point of view the intervening factor may be long Covid, but it may also be many other factors, for this reason I suggest that if you do not have sufficient evidence that long Covid can produce this situation you can include in this paragraph long Covid and other factors, or if you think that long Covid is an important factor you should include information on the prevalence of long Covid in England at least as a justification for your comment.

Response:

We thank the reviewer for their positive feedback and appreciate their suggestion to clarify the role of Long Covid in explaining the model’s performance.

Actions:

We have revised the two paragraphs starting on lines 610 to emphasise that Long Covid is one potential factor among others, and have included relevant information on Long Covid prevalence in England. The text is shown below with additions in red:

“The regression model explaining COVID-19 sickness absence between July 2020 and December 2021 best combines these two streams (ONS estimated positivity, new admissions) into a multivariate model and can explain most of the variability in COVID-19-related sickness absence rates between March 2020 and December 2021. Compared to the univariate ones, the strong performance of this model suggests that each stream contains independent information about the sources of infection for NHS staff (pathways NHS staff can catch infection). These are infection pressure from the community and additional infection pressure from working in hospitals.

The multivariate model combining admission and ONS estimated positivity estimates COVID-19-related sickness absence between July 2020 and early 2022 well before significantly overestimating March-May. This suggests that other dynamics emerged that a statistical model with constant parameters could not capture, for example, the rise in milder but more transmissible sub-variants of COVID-19 in late 2021 [32], as well as changes in policy or the perception of COVID-19 risk. Furthermore, the regression coefficients hide changes in the dynamics. This is evident in the univariate model of hospitalisations explaining the trend between July 2020 and 2021 well, but not afterwards. Another factor that may have weakened the regression models' performance into late 2021 is Long Covid, also known as post-acute COVID-19 syndrome [33]. A subset of individuals infected with COVID-19 experience these prolonged symptoms, including fatigue, breathlessness, and cognitive impairment, which can impact their ability to work [34,35]. ONS data estimates that approximately 1 in 10 individuals infected with COVID-19 in the UK experienced symptoms persisting for 12 weeks [36], with 2.8% of the UK population self-reporting symptoms in April 2022 [37]. These individuals may not have been captured by the COVID-19 surveillance data streams while they remained absent from work. Long Covid may also help explain the overall increased sickness absence rates since 2020. Cases may have been recorded under our COVID-19-related sickness absence category (which includes S13 Cold, Cough, Flu; S15 Chest & Respiratory Problems; and S27 Infectious Diseases). Alternatively, since Long Covid has been linked to persistent psychological and neurological effects [38] [39,40], it may have contributed to the increases in mental health-related sickness absence.”

Reviewer #2

Comment 1:

I have learned that the author used the SARIMA model. But it is not mentioned what type and how many parameters you considered for the mental health-related absence?

Response:

We thank the reviewer for their helpful comment. We performed an exhaustive search across all candidate SARIMA(p,d,q)(P,D,Q)[m] model configurations using the auto.arima function from the R “timetk” package. Our modelling approach is described in the Methods (lines 218 onwards, see below). The fitted SARIMA model structures are shown in the Results section (lines 446, 477) and presented in full in Multimedia Appendix 1.

Actions:

For additional clarity, we edited the paragraph starting on line 218 of the Methods section. The paragraph is shown below with additions in red:

“We partitioned our temporal data into different phases of the COVID-19 pandemic and performed a residual analysis, to identify shifts and anomalies in absence patterns. Specifically, we fit one SARIMA model trained on pre-COVID-19 (January 2015-March 2019) data and used it to predict the post-COVID-19 trend. Additionally, we fit a second SARIMA model including some post-COVID-19 months (January 2015-December 2021). We then forecast the absence rates for the first six months of 2022 using these models and compared the estimates to the first six months of observed data. Model parameters were selected to minimise the Akaike Information Criterion (AIC), using the auto.arima function from R’s “timetk” package [20], with stepwise = FALSE to perform an exhaustive search across all candidate SARIMA(p,d,q)(P,D,Q)[m] model configurations. The resulting fitted models were SARIMA(0,1,3)(0,1,0)[12] for the model using pre-COVID-19 period data for training and SARIMA(2,0,0)(0,1,1)[12] with drift for the model including some post-COVID-19 months. The equations and parameter estimates for both models are given in Multimedia Appendix 1: SARIMA model equations.”

Comment 2:

This study demonstrated that the mental-related absence is significant than S13, S15, and S27. Is this study able to find the key factors that increase the number of absences for mental health issues? At the same time, what is a suggestion for preventing mental health-related absences?

Response:

We thank the reviewer for their suggestions. Identifying the key drivers of mental health-related absences would require a different modelling approach, incorporating additional covariates and more granular data on workers that we do not have access to. Additionally, our study cannot provide evidence-based suggestions for preventing mental health-related absences. However, our results do highlight the importance of continuous mental health support for NHS workers.

Actions:

We have edited the Discussion on lines 705 onwards to highlight both these things, as shown by the red text below.

“Our study shows an increasing trend of mental health-related sickness absence among NHS healthcare workers, highlighting the importance of continued investment in mental health support for workers. However, our time series analysis approach is unable to explicitly identify the driving factors behind this long-term trend. Additionally, we have not explicitly linked indicators of COVID-19 activity to mental health-related absence. How to define the relationship and incorporate a shock function to reflect the pressure on NHS staff is unclear. However, a first step for future investigation could involve incorporating COVID-19 activity indicators as regressors in a time series model, thus providing further insight into the relationship between COVID-19 and mental health-related absences. Moreover, identifying the driving factors behind the long-term mental health-related absences using individual worker-level covariates is a key area for future research.”

Comment 3:

Why is this S10 (mental health-related absence) exponentially increased from March 17, whereas COVID-19 started in December 2019, as shown in Figure 2, panel (a)?

Response:

We thank the reviewer for their observation. The trend suggests longer-term factors influencing mental health-related absences in the NHS workforce. However, our study focuses instead on the impact of COVID-19 on the existing trend. Our SARIMA models (extrapolating from the increasing trend from 2017) highlight that there was a spike in mental health-related absence in early 2020, and suggest that the long-term increase may have been slowing in 2022. These results are discussed on lines 517- 542 (and copied below).

Identifying the factors behind the longer-term trend is beyond the scope of our study and would require a different modelling approach with more granular data. This would be a key area for future research, which we now acknowledge in the text - see our response to Comment 2 above.

Actions:

We added additional text (red below) to the Discussion on lines 517-542 to further highlight the trend and our paper’s focus:

“The data showed that the average rate of mental health-related absences increased consistently year on year between 2017 and 2022. However, our modelling results indicate that there was a spike in mental health-related absence in early 2020, and suggest that the long-term increase may have been slowing in 2022. COVID-19 may have caused a sudden surge or shock in mental health-related sickness absence among NHS staff during the first wave of the pandemic (e.g., possibly stress/anxiety or burnout related). We observed a large deviation from the expected trend according to our time series models, which extrapolated pre-March 2020 data, in April and May 2020. Apart from this period, the deterministic model, characterised by an increasing trend with bi-annual peaks, effectively accounted for the overall trend between June 2020 and November 20212. In addition, Van der Plaat et al. [12] demonstrated a regional correlation between the increase in mental health sickness absence in March-April 2020, compared to 2019, and the cumulative prevalence of COVID-19-related sickness absence during the same period. The authors hypothesized that the spike in mental health-related sickness absence was driven by the combined stresses experienced by healthcare workers, both at work and in their personal lives, as a consequence of the epidemic.

The time series model overestimates the mental health-related absence rate between January and September 2022, which suggests a shift in the underlying dynamics. A possible explanation is that the continuous rise in mental health absences from 2016 was beginning to plateau by 2020. Then, the arrival and constant pressure from COVID-19 had knock-on effects on the well-being of staff and caused a significant increase in mental health absences, but these began to alleviate in 2022. Alternatively, the underestimation could be explained by another reason, such as changes in NHS internal policy regarding absences in this category.”

Comment 4:

What is the limitation of this work and the scope of future study, which is not mentioned here?

Response:

Thank you for highlighting this. We believe the limitations and scope for future work are addressed in the final section of the paper (lines 631-706), where we discuss several limitations and opportunities for future study.

Specifically, we note that while our multivariate model captures COVID-19-related sickness absence well during much of the pandemic, its performance declines from 2022 onwards. This suggests changing dynamics which a model with fixed parameters may not be able to capture, such as the emergence of milder but more transmissible variants, changes in policy and behaviour, and the onset of Long Covid. We also acknowledge that the linear regression models do not capture the underlying mechanisms of COVID-19-related absences.

We also identify several key areas that were not addressed in our current framework, including the role of COVID-19 vaccination, local epidemiological variation, seasonal respiratory viruses, and individual-level and socioeconomic factors - all of which could improve the accuracy and granularity of future models. Additionally, while we observe a rise in mental health-related absence, our analysis does not directly link these trends to COVID-19 indicators, nor does it explain the longer-term drivers of this increase. We propose that future work explore such links, potentially incorporating COVID-19 indicators into time series models of mental health-related absence and using individual-level data to identify underlying causes.

We hope these changes address the reviewers’ concerns and improve the clarity and strength of the manuscript. We are grateful for the opportunity to revise the paper and appreciate the reviewers’ contributions.

Sincerely,

Ewan McTaggart

---

## [Decision Letter · Decision Letter 1]

Sickness Absence Rates in NHS England Staff during the COVID-19 Pandemic: insights from multivariate regression and time series modelling

PONE-D-25-16834R1

Dear Dr. McTaggart,

We’re pleased to inform you that your manuscript has been judged scientifically suitable for publication and will be formally accepted for publication once it meets all outstanding technical requirements.

Kind regards,

Md. Kamrujjaman, Ph.D

Academic Editor

PLOS ONE

Additional Editor Comments (optional):

Reviewers' comments:

Reviewer's Responses to Questions

**Comments to the Author**

Reviewer #1: All comments have been addressed

Reviewer #2: All comments have been addressed

2. Is the manuscript technically sound, and do the data support the conclusions?

Reviewer #1: Yes

Reviewer #2: Yes

3. Has the statistical analysis been performed appropriately and rigorously?

Reviewer #1: Yes

Reviewer #2: Yes

4. Have the authors made all data underlying the findings in their manuscript fully available?

Reviewer #1: Yes

Reviewer #2: Yes

5. Is the manuscript presented in an intelligible fashion and written in standard English?

Reviewer #1: Yes

Reviewer #2: Yes

Reviewer #1: Congratulations, I believe that this manuscriptSickness Absence Rates in NHS England Staff during the COVID-19 Pandemic: insights from multivariate regression and time series modellin, can now be published.

Reviewer #2: The author carefully addressed all questions. I recommend accepting this paper to publish in PLOS ONE.

**Do you want your identity to be public for this peer review?** For information about this choice, including consent withdrawal, please see our Privacy Policy

Reviewer #1: **Yes: ** Dante R Culqui Lévano

Reviewer #2: **Yes: ** DR. MD ANOWAR HOSSAIN

---

## [Editor Report · Acceptance letter]

PONE-D-25-16834R1

PLOS ONE

Dear Dr. McTaggart,

I'm pleased to inform you that your manuscript has been deemed suitable for publication in PLOS ONE. Congratulations! Your manuscript is now being handed over to our production team.

Kind regards,

on behalf of

Dr. Md. Kamrujjaman

Academic Editor

PLOS ONE